# Association between Brain White Matter Lesions and Disease Activity in HAM/TSP Patients

**Keiko Tamaki, Shinji Ouma, Nobutaka Takahashi, Shinsuke Fujioka and Yoshio Tsuboi \***

Department of Neurology, Fukuoka University School of Medicine, Fukuoka 814-0180, Japan;
cerisier.pommier.kei@gmail.com (K.T.); oumas@fukuoka-u.ac.jp (S.O.); ntduel55@gmail.com (N.T.);
shinsuke@cis.fukuoka-u.ac.jp (S.F.)
**\*** Correspondence: tsuboi@cis.fukuoka-u.ac.jp; Tel.: +81-92-801-1011; Fax: +81-92-865-7900

**Abstract:** Human T-cell leukemia virus type 1-associated myelopathy/tropical spastic paraparesis (HAM/TSP) patients may have brain white matter (WM) lesions, but the association of these lesions with disease activity is poorly understood. We retrospectively evaluated the brain WM lesions of 22 HAM/TSP patients (male 4: female 18) including 5 rapid progressors, 16 slow progressors, and 1 very slow progressor. The severity of WM brain lesions on axial Fluid Attenuated Inversion Recovery images was evaluated utilizing the Fazekas scale, cerebrospinal fluid biomarkers, and proviral load in peripheral blood mononuclear cells. Imaging and biological data were compared at the first visit and a subsequent visit more than 4 years later. Patients with comorbidities including adult T-cell leukemia–lymphoma and cerebrovascular disease were excluded. The results revealed that brain WM lesions in the rapid progressors group were more pronounced than those in slow progressors. In patients with HAM/TSP, severe and persistent inflammation of the spinal cord may cause brain WM lesions.

**Keywords:** HAM/TSP; disease activity; brain WM lesions; Fazekas scale

## 1. Introduction

The prevalence of human T-cell leukemia virus type 1 (HTLV-1) infection is estimated to be at least 5–10 million worldwide [1]. The prevalence in Japan is estimated to be at least 1.08 million according to a 2006–2007 survey [2]. Although the majority of HTLV-1-infected individuals remain asymptomatic for life, approximately 3–5% of them develop adult T-cell leukemia–lymphoma (ATL) [3], and 0.25–3.8% develop HTLV-1-associated myelopathy/tropical spastic paraparesis (HAM/TSP) [4]. The clinical course of HAM/TSP is diverse, and it can be divided into three groups depending on the progression. Rapid progressors are those who develop Osame motor disability score (OMDS) [5] grade 5 [needs a cane (unilateral support) to walk] or above within 2 years from the onset of motor symptoms. Very slow progressors are those who progress to OMDS grade 3 (unable to run) or less at least 10 years after the onset of motor symptoms; meanwhile, slow progressors are those who do not meet the definition of either rapid or very slow progressors [6].

Regarding criteria based on the cerebrospinal fluid (CSF) biomarkers, neopterin and C-X-C motif chemokine ligand 10 (CXCL10), at onset in rapid progressors, these biomarkers were found to be significantly higher than in the other two groups: CSF neopterin $\geq 44$ pmol/mL and CSF CXCL10 $\geq 4400$ pg/mL [6]. HTLV-1 proviral load (PVL) in peripheral blood mononuclear cells (PBMCs) has not produced statistically significant differences between the slow and very slow progressors but has shown significantly higher levels in rapid progressors compared to slow and very slow progressors [6,7]. The CSF anti-HTLV-1 antibody (Ab) titer showed statistically significant differences among rapid, slow and very slow progressors, but asymptomatic carriers in the control group tested negative [6,8].

Although it is already known that HAM/TSP patients have brain white matter (WM) lesions [9–11], a difference in incidence between rapid and slow progressors has not been

revealed. We here investigated the association between brain WM lesions and disease progression using blood and CSF findings to indicate disease activity.

## 2. Materials and Methods

### 2.1. Ethical Considerations

This study was approved by the Institutional Review Board of Fukuoka University Hospital (U20-06-002). Prior to the collection of blood or CSF samples and brain magnetic resonance imaging (MRI) examinations, all participants provided written informed consent for analysis of their samples and the results of brain MRI for research purposes as part of their clinical care.

### 2.2. Study Design and Participants

We included 22 patients who were diagnosed with HAM/TSP based on World Health Organization criteria (Osame, 1990) [12] at our department from 2012 to 2022. Clinical information was retrospectively obtained from medical records. Among them, 5 patients were rapid progressors, 16 were slow progressors, and 1 was a very slow progressor (Table 1). Patients who developed ATL and patients with cerebrovascular disease were excluded. None of the 22 patients had cognitive impairment. A family history of HAM/TSP was found in patients 7, 10, 11, 16, and 20, while transfusion history was found in patients 12 and 17. No patient had a history of organ transplantation. The patients were divided into 3 groups based on medical history and CSF biomarkers. Among rapid progressors, patients 3 and 4 developed OMDS grade 5 within 2 years from the onset of motor symptoms. Patient 12 in the slow progressor group was wheelchair-bound due to a traffic accident before the first visit to our hospital. Patient 16 was admitted for the first time at the acute exacerbation stage (OMDS 6→9) with high CSF biomarkers. All patients' demographic data are listed in Table 1.

**Table 1.** All patients' demographic data.

| Patient No. | Sex | Onset Age | Age at Exam | DD | DPP | Brain MRI PVH Fazekas Scale | Brain MRI DWMH Fazekas Scale | PVL in PBMC (Copy/100 Cells) | Anti HTLV-1 Ab Titer in CSF (PA Method) | CSF CXCL10 (pg/mL) | CSF Neopterin (pmol/mL) | Steroid Therapy | OMDS |
|---|---|---|---|---|---|---|---|---|---|---|---|---|---|
| 1 | F | 71 | 71 | 5 m | rapid | 1 | 1 | — | 2048 | 26,772.2 | 63 | no | 9 |
| | | | 71 | 6 m | | — | — | 4.30 | 256 | 1056.2 | 17 | yes | 7 |
| | | | 81 | 11 y | | 3 | 3 | 14.98 | — | — | — | yes | 10 |
| 2 | F | 61 | 61 | 3 m | rapid | 2 | 2 | — | 2048 | 14,492.8 | 70 | no | 7 |
| | | | 67 | 7 y | | 3 | 3 | 3.89 | 1024 | 5083.7 | 43 | yes | 6 |
| 3 | F | 70 | 72 | 2 y | rapid | 1 | 1 | 0.70 | 128 | 1137.2 | 65 | no | 5 |
| | | | 80 | 10 y | | 2 | 1 | 1.69 | 64 | 577.0 | 8 | yes | 6 |
| 4 | F | 65 | 78 | 13 y | rapid | 3 | 3 | 18.69 | 2048 | 7065.4 | 51 | yes | 5 |
| | | | 79 | 14 y | | — | — | 17.55 | — | — | — | yes | 5 |
| 5 | F | 81 | 81 | 3 m | rapid | 1 | 1 | — | 1024 | 3173.6 | 20 | no | 9 |
| | | | 81 | 4 m | | — | — | — | 512 | 485.4 | 9 | yes | 5 |
| | | | 85 | 4 y | | 1 | 1 | — | — | — | — | yes | 5 |
| 6 | F | 58 | 59 | 1 y | slow | 1 | 1 | 13.29 | 128 | 6775.1 | 60 | no | 3 |
| | | | 60 | 2 y | | 1 | 1 | 7.88 | 64 | 4072.4 | 41 | yes | 4 |
| | | | 63 | 5 y | | 1 | 2 | 8.41 | — | — | — | yes | 4 |
| 7 | F | 33 | 48 | 15 y | slow | — | — | 7.97 | 8 | 4463.2 | 20 | no | 4 |
| | | | 53 | 20 y | | 1 | 1 | 12.58 | 8 | 3192.8 | 16 | yes | 5 |
| 8 | F | 59 | 74 | 15 y | slow | 1 | 1 | 8.13 | 64 | 1605.7 | 13 | no | 5 |
| | | | 78 | 19 y | | 1 | 1 | 9.10 | — | — | — | yes | 6 |
| 9 | F | 48 | 61 | 13 y | slow | — | — | 3.09 | 128 | 888.2 | 4 | no | 5 |
| | | | 62 | 14 y | | 2 | 2 | 1.54 | — | — | — | no | 5 |

**Table 1.** *Cont.*

| Patient No. | Sex | Onset Age | Age at Exam | DD | DPP | Brain MRI PVH Fazekas Scale | Brain MRI DWMH Fazekas Scale | PVL in PBMC (Copy/100 Cells) | Anti HTLV-1 Ab Titer in CSF (PA Method) | CSF CXCL10 (pg/mL) | CSF Neopterin (pmol/mL) | Steroid Therapy | OMDS |
|---|---|---|---|---|---|---|---|---|---|---|---|---|---|
| 10 | F | 49 | 69 | 20 y | slow | 0 | 0 | 1.46 | 16 | 82.1 | 3 | no | 5 |
|  |  |  | 76 | 27 y |  | 0 | 0 | 1.76 | — | — | — | no | 6 |
| 11 | M | 40 | 66 | 26 y | slow | — | — | 1.64 | 64 | 784.6 | 5 | no | 5 |
|  |  |  | 71 | 31 y |  | 2 | 2 | 5.79 | — | — | — | no | 6 |
| 12 | F | 68 | 79 | 11 y | slow | — | — | 0.22 | 8 | 220.7 | 4 | no | 9 |
|  |  |  | 83 | 15 y |  | 2 | 2 | 0.42 | — | — | — | no | 9 |
| 13 | F | 27 | 60 | 33 y | slow | 3 | 3 | — | 128 | 1499.1 | 9 | no | 13 |
|  |  |  | 67 | 40 y |  | 3 | 3 | 4.96 | 32 | 467.8 | 6 | yes | 13 |
| 14 | F | 40 | 44 | 4 y | slow | 0 | 0 | 2.77 | 64 | 7499.5 | 32 | no | 4 |
|  |  |  | 52 | 12 y |  | 0 | 0 | 1.79 | — | — | — | no | 5 |
| 15 | F | 55 | 57 | 2 y | slow | 0 | 0 | 2.02 | 128 | 4372.0 | 32 | no | 3 |
|  |  |  | 63 | 8 y |  | 0 | 0 | 3.06 | — | — | — | yes | 4 |
| 16 | F | 10 | 28 | 18 y | slow | 0 | 0 | 6.37 | 2048 | 9107.8 | 49 | no | 9 |
|  |  |  | 28 | 18 y |  | — | — | — | 2048 | 3381.1 | 19 | yes | 6 |
|  |  |  | 33 | 23 y |  | 0 | 0 | 4.88 | — | — | — | yes | 6 |
| 17 | F | 34 | 44 | 10 y | slow | 0 | 1 | 8.87 | 256 | 3576.9 | 15 | yes | 6 |
| 18 | M | 51 | 56 | 5 y | very slow | 0 | 0 | 2.07 | 128 | 458.2 | 4 | no | 3 |
|  |  |  | 63 | 12 y |  | 0 | 0 | 0.81 | — | — | — | no | 3 |
| 19 | M | 54 | 56 | 2 y | slow | 0 | 0 | 8.39 | 32 | 5841.1 | 15 | yes | 4 |
|  |  |  | 61 | 7 y |  | 0 | 0 | 4.04 | — | — | — | yes | 4 |
| 20 | M | 61 | 65 | 4 y | slow | 0 | 0 | 14.59 | 32 | 519.3 | 3 | no | 3 |
| 21 | F | 63 | 66 | 3 y | slow | — | — | 3.84 | 64 | 1464.9 | 10 | no | 4 |
|  |  |  | 71 | 8 y |  | 1 | 1 | 2.98 | — | — | — | no | 6 |
| 22 | F | 67 | 68 | 1 y | slow | 2 | 1 | 1.31 | 64 | 2651.7 | 15 | no | 3 |

—, not tested; CSF, cerebrospinal fluid; CXCL10, C-X-C motif chemokine ligand 10; DD, disease duration; DPP, disease progression pattern; DWMH, deep white matter hyperintensities; F, female; M, male; m, months; OMDS, Osame motor disability score; PVH, periventricular hyperintensities; PVL in PBMC, proviral load in peripheral blood mononuclear cells; y, years.

### 2.3. Magnetic Resonance Imaging

MRI was performed on a 1.5-T scanner (Philips Ingenia from Amsterdam city, Netherlands). We performed both cervical-thoracic spine and brain MRI examinations at the first visit. Brain MRIs, evaluated 4–10 years later, were compared to those captured at the first visit. Brain WM lesions on axial Fluid Attenuated Inversion Recovery (FLAIR) images were evaluated for periventricular hyperintensities (PVH) and deep white matter hyperintensities (DWMH) using the Fazekas scale [13]. Grading of the Fazekas scale is as follows for PVH: 0, no lesions; 1, caps or thin line; 2, smooth halo; 3, extension into the white matter; and it is graded as follows for DWMH: 0, no lesions; 1, punctate foci; 2, beginning of confluence of foci; 3, large confluent areas.

### 2.4. Classification Based on Disease Progression

Motor disability was evaluated using OMDS (Table 2). Patients whose clinical progression to OMDS grade 5 or greater within 2 years after the onset of motor symptoms were diagnosed as rapid progressors, and patients who progressed to OMDS grade 3 or less at least 10 years after the onset of motor symptoms were diagnosed as very slow progressors, while patients who did not meet the definition of either rapid or very slow progressors were diagnosed slow progressors [6].

**Table 2.** Osame motor disability score [5].

| Grade | Motor Disability |
|-------|------------------|
| 0 | No walking or running abnormalities |
| 1 | Normal gait but runs slowly |
| 2 | Abnormal gait (stumbling, stiffness) |
| 3 | Unable to run |
| 4 | Needs handrail to climb stairs |
| 5 | Needs a cane (unilateral support) to walk |
| 6 | Needs bilateral support to walk |
| 7 | Can walk 5–10 m with bilateral support |
| 8 | Can walk 1–5 m with bilateral support |
| 9 | Cannot walk, but able to crawl |
| 10 | Cannot crawl, but able to move using arms |
| 11 | Cannot move around, but able to turn over in bed |
| 12 | Cannot turn over in bed |
| 13 | Cannot even move toes |

*2.5. CSF Biomarkers and PVL in PBMC*

Several CSF biomarkers including neopterin and CXCL10 and HTLV-1 Ab titer were measured. For rapid progressors, 3 out of 5 patients had their CSF biomarkers measured at the time of onset; however, for slow progressors, the first evaluation was performed several decades after onset in some patients, and therefore, CSF biomarker evaluation at the visit more than 4 years later was not performed. The PVL of PBMCs was evaluated at the time of the initial consultation and after more than 4 years, concurrently with the brainMRI examination.

*2.6. Statistical Analysis*

Data related to PVH and DWMH scales as well as CSF biomarkers and PVL in PBMC were analyzed using t-tests between the 2 groups: 5 rapid progressors and 17 slow progressors (including 1 very slow progressor). As this was a retrospective study, some data are missing and therefore not included in analyses. Regarding slow progressors and one very slow progressor, there were patients who had been diagnosed several decades after onset at the time of their first visit to our department, so there is a possibility that the base-point data may not reflect the patient's early onset condition.

In addition to *t*-tests, Pearson's correlation coefficient was used to examine correlations between Fazekas scale of PVH, DWMH, and age at the last point; correlations between Fazekas scale of PVH, DWMH at the last point, and CXCL10 and neopterin of CSF biomarker at the base point; and also the correlation between Fazekas scale of PVH, DWMH, and PVL in PBMC at the last point. Among the 5 rapid progressors, 3 patients did not have a PVL in PBMC measurement at base point; therefore, we used PVL in PBMC values at the last point. Pearson's correlation coefficient was performed using the software package SPSS, v26.0 for Windows (SPSS Inc., New York city, NY, USA). A value of $p < 0.05$ was considered statistically significant.

**3. Results**

*3.1. Fazekas Scale of Brain WM Lesions*

Comparison between the two groups using Fazekas scale data at the last examination point of brain WM lesions showed a significant difference in both PVH ($p = 0.0061$) and DWMH ($p = 0.0217$). Regarding Fazekas scale change for PVH and DWMH between the two points, there was a significant difference only in PVH ($p = 0.0025$) between the two groups.

*3.2. CSF Biomarkers*

CXCL10 and neopterin at the base point showed a significant difference between the two groups at $p = 0.0071$ and $p = 0.0001$, respectively.

*3.3. PVL in PBMC*

PVL in PBMC at the last point did not show a significant difference between the two groups.

*3.4. Correlation between Fazekas Scale of Brain WM Lesions and Age at the Last Point, CSF Biomarkers at the Base Point, and PVL in PBMC at the Last Point*

Pearson's correlation coefficient was used to examine the data. There was a correlation ($r = 0.5$) between age and PVH at the last point. Regarding the CSF biomarkers, there was a correlation ($r = 0.5, 0.5$) between CXCL10 at the base point and PVH and DWMH at the last point and also a correlation ($r = 0.4, 0.4$) between neopterin at the base point and PVH and DWMH at the last point. However, regarding the PVL in PBMC, there was no correlation between PVL in PBMC or PVH and DWMH at the last point. Correlation results were suggestive between PVH of brain WM lesions and age. Furthermore, there were correlations between brain WM lesions and CSF biomarkers CXCL10 and neopterin. Multiple regression analysis of confounding factors (age and disease duration) showed that age affected only PVH, but disease duration had no effect on either PVH or DWMH.

**4. Discussion**

In this study, we evaluated brain WM lesions in patients with HAM/TSP. A final Fazekas scale 3 was more frequent in rapid progressors; conversely, among the group including slow and very slow progressors, only one had Fazekas scale 3. When the slow progressor who scored Fazekas scale 3 visited our department for the first time, it was already 33 years after onset, and her symptoms were severe with OMDS 13. Her cervical and thoracic spine MRI showed severe spinal cord atrophy, especially at the thoracic level. Of the five rapid progressors, three patients eventually presented with Fazekas scale 3 (Figure 1), and they had higher titers of CSF biomarkers (anti-HTLV-1 Ab titer, neopterin, CXCL10) than the other two patients. Regarding PVL in PBMC, there was a patient (patient 2) whose PVL was low among the rapid progressors with Fazekas scale 3. Furthermore, there were no correlations using Pearson's correlation coefficient between Fazekas scale of PVH, DWMH, and PVL in PBMC at the last point in all patients.

We made a scatter plot (Figure 2) between the Fazekas scale of brain WM lesions (PVH, DWMH) at the last point and CSF biomarkers (CXCL10, neopterin) at the base point and age at the last point to make the correlation more visually understandable.

A previous study [14] comparing brain WM lesions between 20 HTLV-1 carriers and 10 HAM/TSP patients shows that brain WM lesion volume was not correlated with OMDS, duration of HAM/TSP, cognitive function, or PVL in PBMC. In another cross-sectional study [15] of 22 HTLV-1 carriers, 22 patients with HAM/TSP, and 18 uninfected controls, brain WM lesions were associated with verbal memory impairment in HAM/TSP patients and carriers, regardless of age, education, or presence of symptoms. Interestingly, there was a correlation between higher PVL in PBMC and neurocognitive dysfunction. In our study, no obvious cognitive impairment was observed even in patients with Fazekas scale 3, and no significant association was found between PVL in PBMC and brain WM lesions. The interrelationships among PVL in PBMC, brain WM lesions, and cognitive function require further discussion in the future. According to another previous study [16] of HAM/TSP patients (6 patients with rapid progression and 22 patients with slow progression), there was no association between brain WM lesions and CSF findings, patient age, or degree of disability. However, there was a significant association between age and periventricular brain WM lesions. In addition, in our study, there was also a correlation between PVH and age at the final time point, suggesting that aging is an important factor for brain WM lesions, but the differences in brain WM lesions according to disease sub-type were significant after adjusting for age. Brain MRI WM lesions in patients with HAM/TSP may reflect chronic perivascular inflammation with progressive gliosis, and autopsy studies of HAM/TSP have shown that inflammatory lesions extend not only to the spinal cord but also to the periventricular region, suggesting brain WM may also be involved in HAM/TSP [17,18].

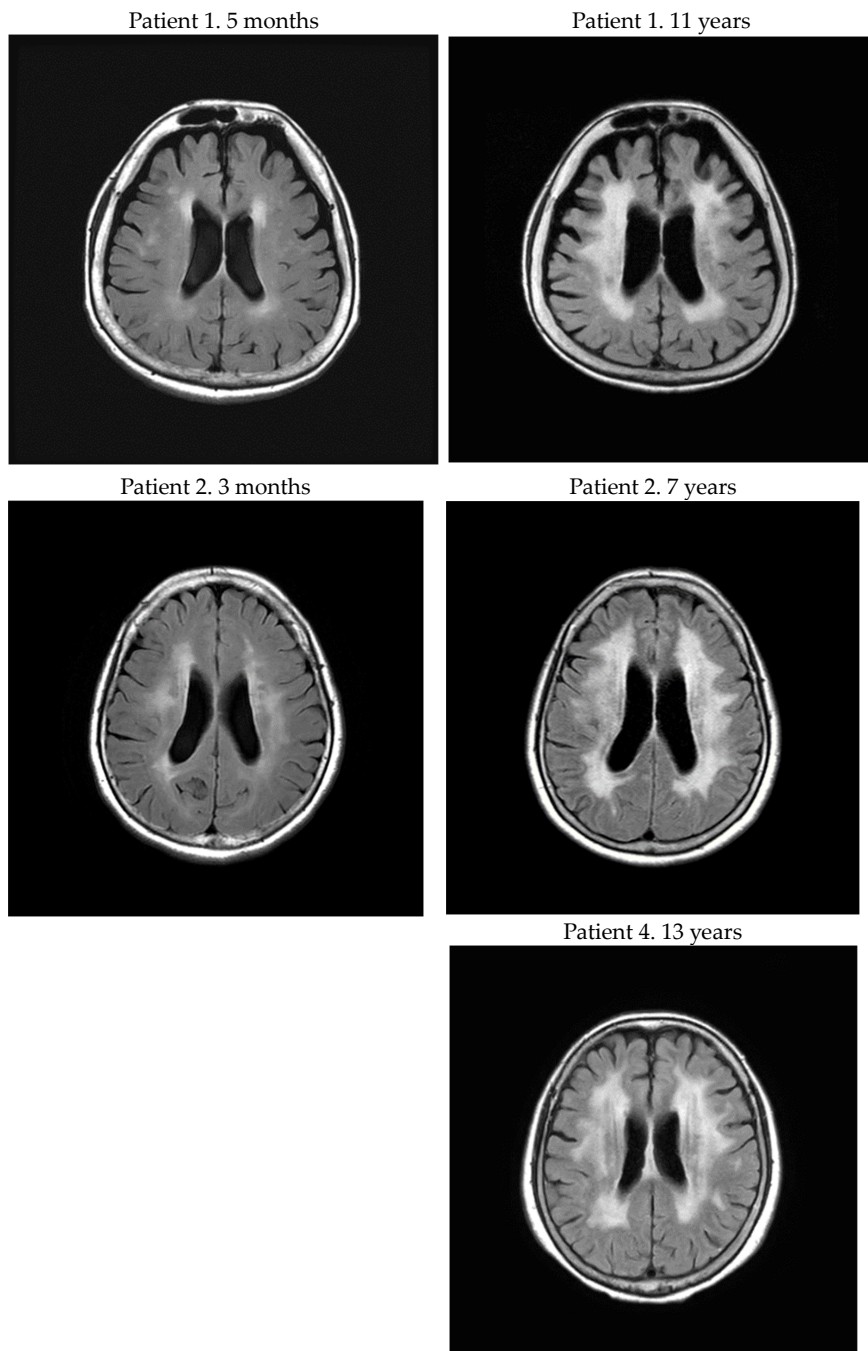

**Figure 1.** Of the 5 rapid progressors, 3 patients eventually presented with Fazekas scale 3, and they had higher CSF biomarkers (anti-HTLV-1 Ab titer, neopterin, and CXCL10) than the other 2 patients.

A strength of our study is that cerebrospinal fluid inflammatory markers were collected continuously, and as a result, cerebrospinal fluid CXCL10 and neopterin were associated with brain WM lesions, independent of age. Patients with rapid progressors in the early stage of the disease are an important factor for poor prognosis, and a correlation between the rate of disease progression and CSF CXCL10 and neopterin levels has been reported [6]. Higher levels of CSF biomarkers may reflect brain white matter and spinal cord damage. Taken together, our results suggested that brain WM lesions are associated with rapid progressors and biological markers such as CSF CXCL10 and neopterin levels. It has also been reported that CSF CXCL10 is useful as a marker for predicting treatment response for HAM/TSP [19].

The limitations of this study are that it was a single-center study and the retrospective data collection had missing data points. The strength of this study lies in the inclusion of

CSF biomarkers (CXCL10 and neopterin) for statistical analysis; through their measurement, the intensity of inflammation in the spinal cord was recorded.

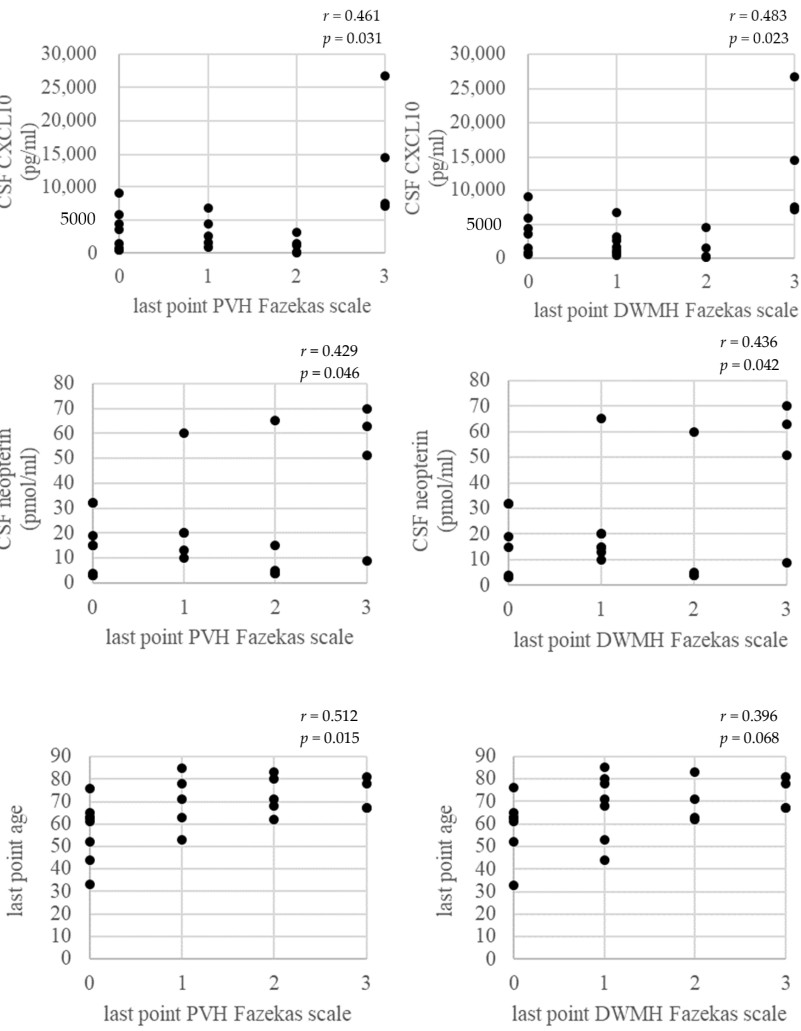

**Figure 2.** Correlation between Fazekas scale of brain WM lesions at the last point and CSF biomarkers at the base point and age at the last point.

Considering these results, it is possible that, in HAM/TSP patients, severe and persistent inflammation of the spinal cord is not confined to the spinal cord, but spreads throughout the neural axis and causes brain WM lesions. A previous necropsy study [18] supports our analysis. Since the number of patients here was small, future large-scale studies are needed.

**Author Contributions:** K.T. conceived and designed the study; K.T., S.O. and N.T. collated the data; K.T. analyzed the data and drafted the manuscript; Y.T. and S.F. critically revised the manuscript. All authors have read and agreed to the published version of the manuscript.

**Funding:** This study was supported by a Health and Labour Sciences Research Grant on Rare and Intractable Diseases from the Ministry of Health, Labour and Welfare of Japan (grant No. JPMH22FC1013).

**Institutional Review Board Statement:** This study was approved by the Institutional Review Board of Fukuoka University Hospital (U20-06-002).

**Informed Consent Statement:** Informed consent was obtained from all patients involved in this study.

**Data Availability Statement:** The data supporting the findings of this study are available on reasonable request from the corresponding author.

**Acknowledgments:** The authors would like to acknowledge the support of Kumi Yamamoto.

**Conflicts of Interest:** The authors declare no conflicts of interest.

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
