# Peer review of "Association between Brain White Matter Lesions and Disease Activity in HAM/TSP Patients"

_2035-8377, doi:10.3390/neurolint16010013_

Round 1

Reviewer 1 Report

Comments and Suggestions for Authors

      I read with interest the short article by Tamaki et al., on the association between brain white matter lesions and disease activity in patients with HAM.

This is a classic retrospective work, quite interesting which concerns a theme where there is still little data available.

It is a shame, on the one hand, that the 3 groups of patients studied are not at all homogeneous in terms of numbers (5, 16 and 1) and, on the other hand, that the different examinations were not carried out homogeneously at the same time (at diagnosis of the disease, and at equivalent dates regarding the evolution of the disease). Additionally, there is a number of missing data. All this is of course linked to the fact that this is a retrospective study based on a series of patients seen over 10 years in a neurology department.

This introduces obvious biases in the results based also on small numbers and therefore a certain difficulty in interpreting data and comparing with other studies on the same theme.

Nevertheless, the results presented in this work, although quite partial, are interesting and informative and deserve to be published.

Author Response

Thank you for your kind and constructive feedback on our research. In future research, we would like to increase the number of cases and avoid missing data.

Reviewer 2 Report

Comments and Suggestions for Authors

Journal: Neurology International

Manuscript ID: neurolint-2802452

Type of manuscript: Brief Report

Title: Association between brain white matter lesions and disease activity in HAM/TSP patients

Authors: Y Tsuboi *, K Tamaki, S Ouma, N Takahashi, S Fujioka 

The authors summarize and report the association between the clinical findings of 22 patients with HTLV-1 infection, including age, inflammation marker in CSF, and white matter lesions on MRI images, and stage progression rate. The information presented in Table 1 is detailed. The paper is also concise and clear. This reviewer recognizes that this research is time consuming due to the unique nature of the subject matter and that it is difficult to obtain similar data on an ongoing basis. This reviewer has determined that the following changes and additions to the text would allow the paper to be published as a brief report in Neurology International.

This reviewer believes that this manuscript adequately answers this reviewer's question, enhances the description of the work, and clarifies the argument.

1. The Results section (L134-160) contains statistical values from the information in Table 1. This reviewer suggests using a bar graph or similar to make the comparison more visually understandable.

2. In the Discussion section, this reviewer would like to see in L179-188 the reason for the difference in results from the cited document [9].

3. Regarding L196-200, there are some leaps in logic, so this reviewer would like the text to be further organized to make it more persuasive. Please summarize the contribution of factors that contribute to the rate of progression of HAM/TSP, and if possible, the causal relationship.

4. As the authors state in L207-209, the strength of this study was the continuous collection of data on the inflammatory response. However, there is little mention of this in the text. This should be discussed in conjunction with point 3.

Author Response

Thank you for your kind and constructive feedback to our retrospective work.

  1. The Results section (L134-160) contains statistical values from the information in Table 1. This reviewer suggests using a bar graph or similar to make the comparison more visually understandable.

Thank you for your suggestion. We made scatter plot between Fazekas scale of brain WM lesions (PVH, DWMH) at the last point and CSF biomarkers (CXCL10, neopterin) at the base point, age at the last point as Figure 2. We added the comment in Line 174-176.

  1. In the Discussion section, this reviewer would like to see in L179-188 the reason for the difference in results from the cited document [9].

Thank you for reviewer’s comments. The results of the cited document [9] are not different from our results. In both studies, brain WM lesions did not correlate with disease duration or PVL in PBMCs. Our results add new findings suggesting that brain WM lesions are associated with rapid progressors and inflammatory markers in the cerebrospinal fluid. The cited document [9] became [14].

  1. Regarding L196-200, there are some leaps in logic, so this reviewer would like the text to be further organized to make it more persuasive. Please summarize the contribution of factors that contribute to the rate of progression of HAM/TSP, and if possible, the causal relationship.
  2. As the authors state in L207-209, the strength of this study was the continuous collection of data on the inflammatory response. However, there is little mention of this in the text. This should be discussed in conjunction with point 3.

We appreciate reviewer’s thoughtful comments and suggestions in 3 and 4.

We have revised the comments made in the discussion based on the reviewer's suggestions, in Line 234-243.

“A strength of our study is that cerebrospinal fluid inflammatory markers were collected continuously, and as a result, cerebrospinal fluid CXCL10 and neopterin were associated with brain WM lesions, independent of age. Patients with rapid progressors in the early stage of the disease are an important factor for poor prognosis, and a correlation between the rate of disease progression and CSF CXCL10 and neopterin levels has been reported [6]. Higher levels of CSF biomarkers may reflect brain white matter and spinal cord damage. Taken together, our results suggested that brain WM lesions are associated with rapid progressors and biological markers such as CSF CXCL10 and neopterin levels. It has also been reported that CSF CXCL10 is useful as a marker for predicting treatment response for HAM/TSP [19].”

Again, we appreciate all your insightful comments. Thank you for taking the time and energy to help us improve the paper.